# Multiple Posterior Insula Projections to the Brainstem Descending Pain Modulatory System

**DOI:** 10.3390/ijms25179185

**Published:** 2024-08-24

**Authors:** Despoina Liang, Charalampos Labrakakis

**Affiliations:** 1Department of Biological Applications and Technology, University of Ioannina, 45110 Ioannina, Greece; 2Institute of Biosciences, University Research Center of Ioannina, 45110 Ioannina, Greece

**Keywords:** anterograde tracing, mCherry, posterior insular cortex, periaqueductal gray, nucleus raphe magnus, parabrachial nucleus, noradrenergic nuclei, LC, A5, spinal cord dorsal horn

## Abstract

The insular cortex is an important hub for sensory and emotional integration. It is one of the areas consistently found activated during pain. While the insular’s connections to the limbic system might play a role in the aversive and emotional component of pain, its connections to the descending pain system might be involved in pain intensity coding. Here, we used anterograde tracing with viral expression of mCherry fluorescent protein, to examine the connectivity of insular axons to different brainstem nuclei involved in the descending modulation of pain in detail. We found extensive connections to the main areas of descending pain control, namely, the periaqueductal gray (PAG) and the raphe magnus (RMg). In addition, we also identified an extensive insular connection to the parabrachial nucleus (PBN). Although not as extensive, we found a consistent axonal input from the insula to different noradrenergic nuclei, the locus coeruleus (LC), the subcoereuleus (SubCD) and the A5 nucleus. These connections emphasize a prominent relation of the insula with the descending pain modulatory system, which reveals an important role of the insula in pain processing through descending pathways.

## 1. Introduction

The top-down control of pain allows for a context-dependent experience of pain, in which the behavioral and emotional state, attention and past experiences modulate the perception of pain. This system is fundamental for the expression of endogenous analgesia and is a main target of the pharmacological treatment of pain by opioids [1]. Descending pain modulation is bidirectional; in addition to descending pain inhibition, a descending facilitatory pathway can enhance pain-related signal transmission in the spinal cord and this pathway might be important in the expression of chronic pain [2,3]. The control center of this descending system lies in the brainstem, giving rise to afferent outputs to the spinal cord that are involved in the processing of the incoming nociceptive messages from the periphery.

The PAG is a major hub in the descending pain modulatory system, operating through its outputs to the rostral ventromedial medulla (RVM) and LC to modulate pain [4,5,6,7,8]. It is a main site for opioid-induced analgesia [9,10]. The PAG receives inputs from several cortical and subcortical regions involved in pain processing. The medial prefrontal cortex (mPFC) input to the PAG has been identified to mediate endogenous descending pain analgesia [11,12,13]. Consequently, reduced mPFC output to PAG is involved in the development of neuropathic pain. Another analgesic input to the PAG is stemming from the orbitofrontal cortex [14]. Additional inputs to the PAG originate in the hypothalamus. Both projections from the lateral hypothalamus [15,16] and the parvocellular hypothalamic nucleus [17] are antinociceptive. On the other hand, inputs from the anterior cingulate cortex are pronociceptive [18,19]. Connections from the amygdala have also been described [20], which have a pro-nociceptive action [21]. Finally, a direct bottom-up connection from secondary nociceptive neurons from the spinal cord [22,23] might directly gate the descending control system. Thus, spinal nociceptive inputs could potentially activate or suppress the pain modulatory circuits of the PAG that ultimately influence pain processing in the spinal cord.

The RVM receives pain modulatory inputs from the PAG [24] and is considered the main output of the descending pain control system to the dorsal horn spinal cord [4]. Three types of cells have been identified: ON, OFF and neutral cells [25]. While OFF cells inhibit nociceptive processing [26], ON cells promote nociception [27]. The RVM also receives inputs from other pain-related brain areas, including the parabrachial nucleus, prefrontal cortex, locus coeruleus and hypothalamus [24,28,29]. Stimulation of the pontine noradrenergic nucleus of LC has also been shown to elicit descending inhibition [30] through its spinal projections [31,32]. The LC receives inputs from the other brainstem nuclei involved in the descending modulation of pain, the PAG and RVM [33,34]. It also receives inputs from other brain areas involved in pain processing, like the lateral hypothalamus [35] and mPFC [36], as well as from other noradrenergic nuclei, i.e., A7 and A5 [32]. 

The insular cortex is one of the brain areas that integrates sensory and emotional information and is found activated during pain in almost all brain imaging studies [37]. It is divided into a more sensory part, the posterior insula; and a more limbic part, the anterior insula [38]. The posterior part of the insula has been proposed to encode pain intensity [39]. In addition, stimulation of the insula produces the sensation of pain. Recent research suggests that the insula might exert its actions through its projections to the descending pain modulatory system [37,40]. Projections from the insula to the PAG, RMg and the LC have previously been shown mainly in rats using traditional tracing techniques [24,40,41,42,43,44] Here, we sought to confirm insular projections to the above nuclei in the mouse, and to consolidate information on the relative projection strength to the different parts of the DPMS from the same set of experiments. In addition, we aimed to broaden our knowledge of the insula DPMS connectivity to other parts of noradrenergic system, like the A5 nucleus, that are implicated in descending control [45,46] but have previously not been explored. Finally, we also explored the presence of direct descending projections form the insula to the spinal cord. We used viral mediated anterograde tracing that allows for cell type specific fluorescent tracer protein (mCherry) expression to glutamatergic pyramidal neurons. Robust projections from the posterior insula to the PAG, the RMg, part of the RVM, and the PBN are shown. In addition, we use tyrosine hydroxylase (TH) immunostaining to identify insula axonal projections to different noradrenergic nuclei and show LC, A5 and SubCD nuclei projections. 

## 2. Results

### 2.1. Injection Sites

Stereotactic injections of AAV constructs led to the expression of mCherry in pyramidal neurons of the right insular cortex (Figure 1B,C). Only animals with the majority of expression within the posterior insular cortex were used in this study (*n* = 4). Six animals were excluded, of which two had no mCherry expression and in the other four, mCherry was expressed majorly in the secondary somatosensory cortex (S2) or the claustrum and the piriform cortex. In the included animals, only some expression was observed in the claustrum in two of the cases. In one case, few cells within the piriform cortex adjacent to the agranular insula were labelled and, in another case, sparse expression in the secondary somatosensory cortex was found. The extend of expression is shown in Figure 1A and Appendix A. The peak of the viral transfection was between bregma −0.65 and −0.94 mm (Appendix A). The center of the injection in two cases was shifted more rostrally, while in the other two cases, a more caudal expression was found (Appendix A).

### 2.2. Axonal Projections from the Insula to the Brain Stem

#### 2.2.1. Periaqueductal Gray

Axons and varicosities expressing mCherry were prominently found within the right ventrolateral PAG (vlPAG; Figure 2A,B and Appendix A) in all injected animals (*n* = 4), showing posterior insular projections ipsilaterally. Projections to the contralateral site were also found, although these were lower in density (Figure 2C,D and Appendix A). Insular projections to the lateral PAG (lPAG) were also present (Appendix A). Projections to the dorsolateral (dlPAG) or dorsomedial PAG were virtually nonexistent or very sparse in most cases except one case (Figure 2G,H and Appendix A). Dopaminergic neurons in the ventral PAG/dorsal raphe nucleus have been previously reported to play a role in pain [47]. We used anti-TH staining to delineate this area and found insular projections within this area too (Figure 2E,F and Appendix A). In order to locate the neuronal cell bodies within the insular cortex that project to the PAG, we injected the fluorescent cholera toxin beta sububunit (CTB) retrograde tracer in the PAG. We found labelled cell bodies within the ipsilateral posterior insular cortex in both the agranular and granular/dysgranular parts (Figure 3).

#### 2.2.2. Raphe Magnus Nucleus

We explored the expression of mCherry in the RMg, a main part of the RVM. Axonal projections and varicosities from the posterior insula immunoreactive for mCherry were found in the RMg of all animals studied (*n* = 4). Projections were found in both the ipsilateral and contralateral sites (Figure 4A–D and Appendix A). Other parts of the RVM also contained axons from the insula (Appendix A) but were not analyzed further in this study.

#### 2.2.3. Locus Coeruleus and Other Noradrenergic Nuclei

To identify noradrenergic brain nuclei, we used immunofluorescence for TH, the rate-limiting enzyme in the catecholamine and noradrenaline biosynthetic pathway. TH staining diffusely labelled cell somata and dendrites, which allowed for the identification of LC, SubCD, A5 and A7 nuclei. Axonal, mCherry labelled projections were found in LC, SubCD and A5, both in the ipsilateral and the contralateral sites (Figure 5A–E) in all mice studied (*n* = 4). In addition, axonal projections were also detected in the A7 nucleus (Figure 5F). In some cases, we were able to identify mCherry labelled varicosities in close contact to TH labelled neurons (Figure 5B), indicating possible synapses between insular neurons and noradrenergic cells.

#### 2.2.4. Parabrachial Nucleus

The lateral parabrachial nucleus (lPBN) receives direct nociceptive inputs from the superficial dorsal horn [48], but it is also involved in the descending pathways [28]. We therefore examined the possibility that the posterior insula projects to the lPBN. Dense mCherry-immunoreactive axonal fibers (Figure 6A) were present throughout the ipsilateral lPBN in all mice studied (*n* = 4), indicating considerable posterior insula to PBN connection. Projections to the contralateral lPBN were also found (Figure 6B). In addition, mCherry immunoreactivity was found in the medial PBN (Appendix A).

#### 2.2.5. Dorsal Horn of the Spinal Cord

Previous data have shown a direct descending functional connection of the insular cortex to the spinal trigeminal nucleus of the medulla [41,49]. We therefore asked the question of whether the posterior insula also sends projections to the dorsal horn of the spinal cord. We were not able to detect mCherry immunofluorescent fibers in the lumbar spinal cord (Figure 7A,C) in all mice studied (*n* = 4). This lack of axonal projection was also confirmed in the thoracic dorsal horn (Figure 7B).

### 2.3. Relative Projection Densities to the Different Brain Stem Areas

To compare the insular projections to the brainstem, we quantified the total axon length density in the ipsilateral vlPAG, RVM, LC, SubCD, A5 and lPBN. The PAG, RVM and lPBN showed the highest axonal densities (Figure 8). The noradrenergic nuclei had axonal projections that were comparable between themselves, indicating consistent projection from the insula, but significantly lower than those of the other regions (Figure 8).

## 3. Discussion

Here, we show multiple projections of posterior insular pyramidal neurons to several brainstem nuclei that play a role in descending pain modulation. We used viral mediated anterograde tracings, that are based on the expression and transport of fluorescent proteins along the axons. Our anterograde tracing study shows a substantial projection from the posterior insula cortex to the central hub of the descending pain pathway, the PAG, in mice. This projection was mainly to the ipsilateral part of the PAG and was found mainly in the l/vlPAG and the dorsal raphe nucleus. Contralateral projecting axons were also found. Previous work, in rats, using biotylinated dextran amine as an anterograde tracer, did show insula stemming axons in l/vlPAG [41]. Interestingly, posterior insula projections were more dense than anterior insula axonal projections. In a different study in rats, Floyd and colleagues [42] used both anterograde and retrograde stainings to identify insula–PAG connectivity. Anterograde labelling showed axonal projections from the anterior insula to the l/vlPAG mainly. The retrograde stainings confirmed anterior and posterior insula projections to the l/vlPAG. A marked difference between the two studies was the sub-insular origin of the PAG axons. While, in the study of Floyd et al. [42], retrogradely labelled somata were found in the agranular regions, the anterior dyes used by Sato et al. [41] were placed in granular and dysgranular insular regions, suggesting a granular origin of the axons. Our retrograde tracing from the PAG shows cell bodies in all posterior insular subregions. Insular projections to the dorsal raphe were not reported, to our knowledge, in the above or other studies before. Similar to Floyd et al. [42], we found occasional axonal labelling in the dorsal PAG, which could be a result of spread of the viral vector to the claustrum.

Considerable axonal projections from the insula were found in the RMg. The RMg is part of the of RVM, receives inputs from the PAG [8] and is an important output region of the descending pain modulatory system to the spinal cord. Indeed, stimulation of RMg serotonergic neurons causes pain hypersensitivity [50]. Insula-to-RMg projections have been described in rats, mainly from retrograde injection to the rostral RMg [24]. In another anterograde study in rats, projections from granular anterior and posterior insula to the RMg have been confirmed [41,43]. Projections from the posterior insula to the RMg were also reported in mice [40]. Our study confirms this projection.

We also show axonal posterior insular projections to brainstem noradrenergic nuclei in the mouse. Several of the noradrenergic nuclei received axons from the insular cortex, including the LC, SubCD, A5 and A7. Sparce axonal projections to the LC have previously been reported from the medial and anterior insula in rats [44]. Projections from the anterior insular cortex to GABAergic neurons near the LC have also been shown in the rat [43]. To our knowledge, insular projections to the SubCD, A5 and A7 nuclei have not been reported before. An advantage of this study is the use of TH immunoreactivity for improved identification of the different nuclei. In addition, it allowed us to identify axonal varicosities in close appositions to TH-positive somata, which possibly represents direct synaptic connections between posterior insula pyramidal neurons and LC noradrenergic neurons. Insular axonal innervation was comparable between the different noradrenergic nuclei, indicating similar roles of insula–noradrenergic system projections. On the other hand, insular projections to the noradrenergic system were almost an order lower than projections to other brainstem nuclei like PAG and RMg. This correlates with the sparse innervation reported in older studies [44]. Lower density might reflect different and/or more specialized roles of the insula–noradrenergic system connection. On the other hand, the lower density might be compensated by the multiple projection to the different noradrenergic nuclei. The LC plays a role in pain and is involved both in analgesia and pain facilitation [51]. Recent research suggest that the LC is organized into modules with diverse roles [31,32]: an ascending module, that lies dorsally and projects to the frontal cortex, and a spinally projecting module, which is located in the ventral part of the LC. On the other hand, the SubCD nucleus extends rostroventral from the LC and also projects to the spinal cord [45]. Thus, the ventral LC with the SubCD might make up a single functional module that projects to the spinal cord, mediating descending inhibition. Interestingly, the SubCD part received enhanced axonal innervation from the insula. Projections to the spinal cord have also been described for the other noradrenergic nuclei of A5 and A7 [45,46] and these nuclei have also been involved in descending pain modulation [52,53,54,55]. Thus, insular projections could be involved in descending pain modulation through the LC and other noradrenergic nuclei. Functional evidence of the insula–LC role in pain was provided for the anterior insula [43], it was shown that GABAergic inhibition of the anterior insular cortex had an antinociceptive effect that was spinal a-adrenoreceptor-dependent.

Interestingly we found a major insular input to the PBN. This input was comparable in density with that of the insula–PAG and the insula–RMg inputs, indicating an important role in the control of PBN function. The PBN plays a role in homeostatic processes and acts as a nociceptive sensory relay. It receives direct inputs from secondary ascending nociceptive neurons from the dorsal spinal cord [56]. A connection of the PBN to the DPMS was shown through its connection to the RVM [28,57]. However, a direct descending action to the medullary dorsal horn has also been suggested [58]. In addition, the dlPBN sends projections to the lPAG [56,59]. Posterior insula to lPBN projections have been previously described in the rat [41]. Axons from the anterior insula in proximity to PBN GABAergic neurons were also shown in the rat [43]. Thus, the posterior insula might be involved in nociceptive processing in the PBN that could affect both ascending and descending information.

For all brainstem areas studied, beside the insular projections to the ipsilateral site, projections were also found to the contralateral site. This correlates with observations of bilateral pain modulation after unilateral stimulation of the insula [60,61]. Thus, contralateral projections could constitute the underlying mechanism for these observations as well for the diffuse nature of the conditioned pain modulation [62].

Because previous research has indicated that the insular cortex projects to the medullary dorsal horn [41] and that it could directly influence pain processing [49], we tested if the posterior insula also sends axons to the spinal part of the dorsal horn. Our experiments failed to show axonal projections to the lumbar or thoracic parts of the spinal cord. Thus, it could be that insula mediates direct descending control for orofacial pain only, while it influences somatic pain sensation indirectly through the canonical DPMS.

Posterior insula stemming axons in the studied DPMS areas displayed axonal varicosities, an indication of the presence of synapses [63]. This implies functional specificity of the presence of the axons in the studied areas as compared to passing axons. The higher occurrence of axonal structures in the studied brainstem areas compared to neighboring structures (Appendix A) corroborates this specificity. Besides the insular projections explored here, another medullary nucleus that receives insular projections is the subnucleus reticularis dorsalis [64], which is also involved with descending pain modulation [65]. Our results and those of previous studies stress the robust connection of the posterior insula with the parts of the brain involved with descending pain modulation, thus indicating an important role of the insula in descending pain control. Experimental evidence for this role of the insula has been shown for the posterior insula to the raphe magnus connection. Activation of this circuit, or its upstream input from the midcingulate cortex, causes pain facilitation in a spinal serotonin-dependent manner [40]. In addition, earlier results provided some evidence for an anterior insula–descending pain system circuit modulating pain responses. This, however, was found to be spinal noradrenaline-sensitive [43]. On the contrary, the studied brainstem regions along with pain modulation, are also involved in diverse other functions, for example arousal, homeostasis, escape and coping behaviors, taste aversion and appetitive processes, and thermoregulatory and other autonomic functions [66,67,68,69,70]. It is therefore plausible that the insula–brainstem connection is involved in multiple functions in addition to pain.

In this study, we used a viral vector approach to trace posterior insula axonal projections to different parts of the DPMS, which offers several advantages. Viral vectors are using the cellular machinery for tracer production. This potentially would allow for the use of a lower concentration of injected viral constructs than that required by traditional dyes. Because mice brains are relatively small, a smaller titer could contain and restrict the spread of transfection within the posterior insula cortex. Indeed, we used low-titer viral injections, ~30× less concentrated than those used in other published work, with minimal spread to neighboring structures. Because the tracer is produced within the cells, the extend and spread of the injection can be better quantified than traditional tracer dyes by counting successfully transfected cells. In addition, the use of viral vectors also allows for cell type specificity in the expression of the tracer. Here, we used the CAMKIIa promoter, which targets glutamatergic pyramidal neurons in the cortex. Thus, we avoid possible mCherry expression in GABAergic neurons. Although most cortical projections are glutamatergic, some inhibitory projections from the insula and other cortical areas are reported in recent studies [71,72]. One caveat of using AAVs as anterograde tracers, including the AAV8 serotype used in this study, is the possibility of some retrograde tropism [73], although this possibility is low [74] and should virtually be non-existent in our low titer experiments. Indeed, we never detected retrogradely labeled neurons in areas that heavily connect to the insula [75], like the sensory cortices, thalamus and amygdala, as well as the contralateral insula (Figure 1B). Likewise, we never detected any mCherry-positive cell-like structures in the DPMS target regions studied.

Further research should shed light on the diversity of functions of these multiple connections of the insula to the brainstem and the descending pain modulatory system. It remains to be answered if these multiple connections serve as a mechanism of redundancy or constitute different aspects of pain processing. Future functional evaluation of the posterior insula to the DPMS circuit could reveal substantial mechanisms involved in pain hypersensitivity that could ultimately be targeted for pain relief [76].

## 4. Materials and Methods

### 4.1. Animals

Experiments were carried out in male CD1 3–5-month-old mice. Animals were housed in groups of 2–4, at a 12 h light/12 h dark cycle and with ad libitum access to food and water. All procedures were in accordance with the European Communities Council Directives 2010/63/EU and were approved by the Prefecture of Epirus and the University of Ioannina animal care and use committee.

### 4.2. Stereotactic Injections

Mice were anesthetized by ketamine (Richter Pharma, Wels, Austria; 100 mg/kg), xylazine (Vetoquinol, Lure, France; 10 mg/kg) and acepromazine (Alfasan, Woerden, The Netherlands; 1 mg/kg) and were placed in a stereotactic rig (RWD life science, Shenzhen, China). The skull was exposed, and a small craniotomy was made on its right side. The adeno-associated viral vector AAV8-CaMKIIa-mCherry (Addgene, Watertown, MA, USA) at a titer of 6.3 × 10^10^ was delivered via a Hamilton Neuros syringe in a volume of 200–300 nl (at a rate of 100 nL/min). The coordinates of the injection were modified from Paxinos and Franklin [77] and were as follows: AP −0.7, DV −3.9 and ML −4.3 mm. The skin was closed with sutures and mice were treated with the analgesic ibuprofen and the antibiotic amoxicillin for 72 h postoperatively. Mobility, eating, drinking and weight were monitored for the 5-day postoperative duration.

For retrograde tracings, cholera toxin B subunit conjugated with CF640 fluorophore (Biotium, Fremont, CA, USA) was injected in the right PAG (0.2%, 200 nL, AP −4.75, DV −3.2, ML −0.4 mm).

### 4.3. Immunofluorescence

At 4–8 weeks after injections, mice were anesthetized by ketamine and xylazine and transcardially perfused with saline followed by 4% PFA. Brains were removed and postfixed overnight. Serial coronal 30 μm sections were made with a vibratome in PBS and slices were plated on slides. Slides were incubated with primary antibodies overnight at 4 °C. The primary antibodies used were chicken anti-mCherry (1:2000; Agrisera, Vännäs, Sweden) and rabbit anti-tyrosine hydroxylase (1:2000; Proteintec and Pel-Freez, Richmond, VA, USA). Secondary antibodies were incubated for 2 h in room temperature. Goat anti-Chicken IgY conjugated to CF640 (1:500; Biotium, Fremont, CA, USA) and goat anti-rabbit IgG conjugated to CF488 (1:500; Biotium, USA). Slides were incubated with DAPI (1 μg/mL; Sigma, St. Louis, MO, USA) and mounted with Vectashield (Sigma, USA).

### 4.4. Imaging and Statistics

Immunofluorescence images were collected with a laser scanning confocal microscope (Nikon AX R, Tokyo, Japan) with identical settings across animals. Identification of brain areas were based on the Paxinos and Franklin atlas [77]. The LC and the other noradrenergic nuclei were delineated using TH immunofluorescence. Analysis of the LC was made at the dorsoventrally elongated part of LC just below the 4th ventricle (Figure 5A; −5.30 to −5.50 μm from Bregma in the Paxinos Atlas). For the SubCD, we used slices 120 μm anterior from the position of the most rostral elongated part of LC, where TH-positive neurons were found at a level 1 mm ventrally from the 4th ventricle (Appendix A; level equivalent to Bregma −5.20/−5.30 μm in Paxinos Atlas). Analysis of the PAG was made from the caudal part (equivalent locations to Paxinos Atlas panels 68–70). Raphe magnus analysis was made from the rostral medullar part (equivalent locations to Paxinos Atlas panels 78–80). Image analysis was performed using the ImageJ software v1.53c (NIH, Bethesda, MD, USA).

The extend of mCherry expression was visually inspected from selected 30 μm sections spanning the rostro-caudal length of the posterior insula at the equivalent levels of −0.2, −0.35, −0.65, −0.95 and −1.10 mm from Bregma [77]. Green autofluorescence was used for anatomical hallmark identification. Spatial extend and relative intensity was arbitrarily estimated and included mCherry expression in cell somata, dendritic and axonal processes. Computer-assisted manual drawings of the spatial extend were converted to images and averaged for the different animals in ImageJ. For the quantitative assessment of viral transfection, mCherry-positive cell somata in the above locations were counted. Oval and pyramidal shaped fluorescent structures enclosing a DAPI stained cell nucleus were identified as somata (Figure 1C and Appendix A).

For axon length density, higher magnification (40×) images of z-scans (25 image stacks, a total of 15 μm thickness) were summed. Axons were identified by their thin, fiber-like structure, with or without branches or varicosities and distinguished mCherry fluorescence. Axons were traced manually in imageJ, and the sum of lengths was calculated and normalized to 1 μm^3^ volume. Only structures with >3 μm length were included. Statistical analysis with one-way repeated measures ANOVA was performed with SPSS software v28.0.1.0 (IBM, New York, NY, USA).

## Figures and Tables

**Figure 1 ijms-25-09185-f001:**
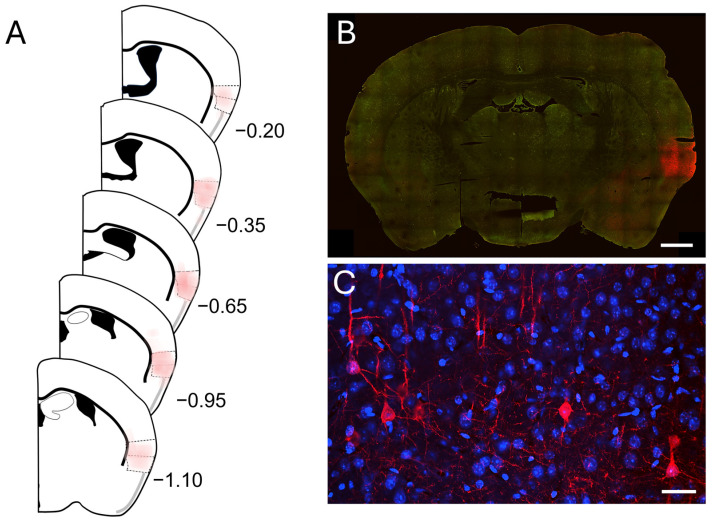
Virus-mediated mCherry expression in the right posterior insular cortex. (**A**) Coronal schematic diagrams showing the extend of mCherry expression at different anterior-posterior levels. The colored shading is the superimposed average of expression from all mice (*n* = 4) used in this study. The bulk of the expression was between levels of −0.3 mm and −1.0 mm from bregma. (**B**) A representative immunostaining of a coronal section for mCherry (red), showing expression within the insular cortex. Green autofluorescence is used for anatomical feature reference (scale bar: 1 mm). (**C**) Higher magnification image of layer 5 posterior insula pyramidal neurons expressing mCherry (red). Nuclear DAPI staining is shown in blue. Immunofluorescent cell somata, dendrites and some axons are visible (scale bar: 30 μm).

**Figure 2 ijms-25-09185-f002:**
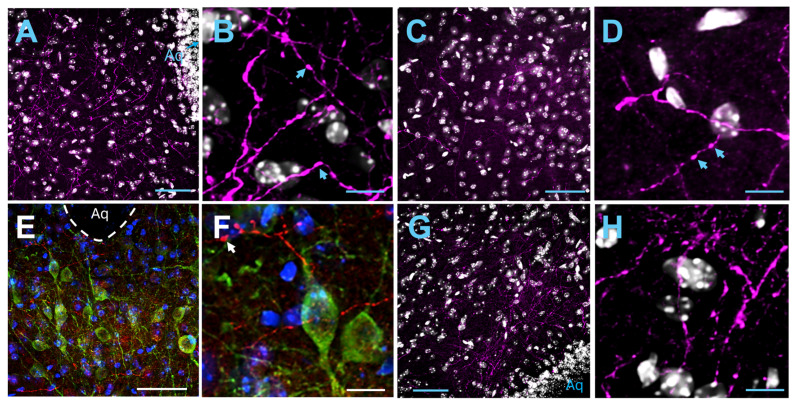
Posterior insular projections to the PAG. (**A**) Axonal projections labeled with immunofluorescence for mCherry (magenta) in the vlPAG ipsilateral to the injected insula. Nuclei are stained with DAPI (grey). (**B**) A higher magnification of the section in (**A**). Insula emanating axons with en-passant varicosities are visible. (**C**) Posterior insula emanating axons (magenta) in the contralateral vlPAG. (**D**) A higher magnification of the section in (**C**). (**E**) Immunofluorescence for mCherry (red), and TH (green) shows posterior insular projections ventral of the Aqueduct (Aq) within the dopamine neuron containing dorsal raphe nucleus. DAPI staining is shown in blue. (**F**) A higher magnification of the section in (**E**). (**G**) Insula axonal projections in the ipsilateral dlPAG. (**H**) A higher magnification of the section in (**G**). Scale bars: 50 μm (**A**,**C**,**E**,**G**), 10 μm (**B**,**D**,**F**,**H**). Arrows denote varicosities.

**Figure 3 ijms-25-09185-f003:**
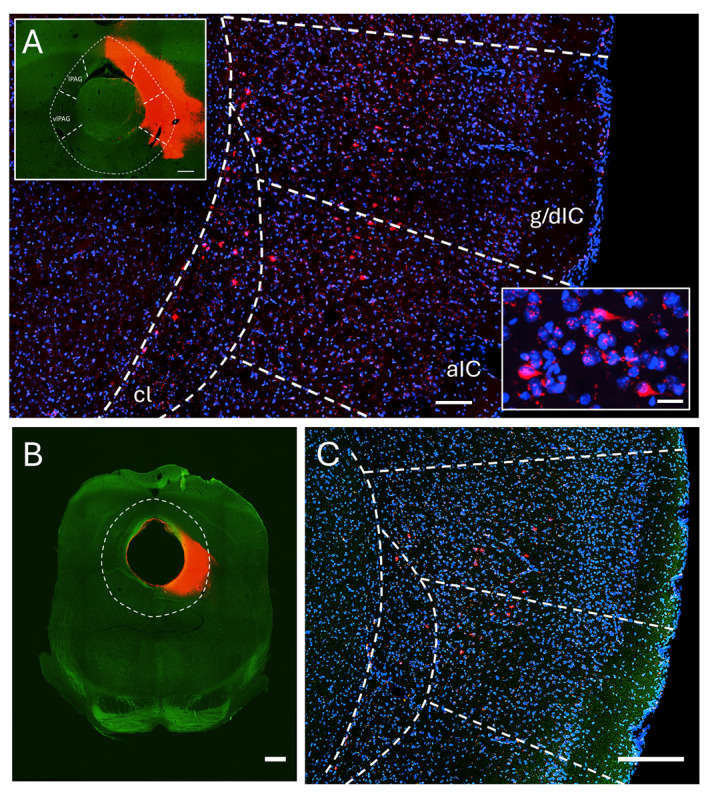
Retrograde labelling of Insula–PAG projections. (**A**) CTB-labeled cell bodies in the posterior insula (red) lie mainly in layers V and VI in both the agranular (aIC) and granular/dysgranular (g/dIC) subareas of the posterior insular cortex (blue: nuclear DAPI staining; cl: claustrum; scale bar 100 μm). Upper inset: The CTB fluorescence (red) at the injection site is shown (green: 488 nm autofluorescence for anatomical reference; scale bar: 300 μm), Lower inset: of CTB labelled somata scale bar: 20 μm. (**B**) CTB (red) injection site from a different animal (dashed line delineates the PAG, scale bar: 500 μm). (**C**) CTB-labeled cell bodies in the posterior insula (red) from the example in (**B**) (scale bar: 250 μm).

**Figure 4 ijms-25-09185-f004:**
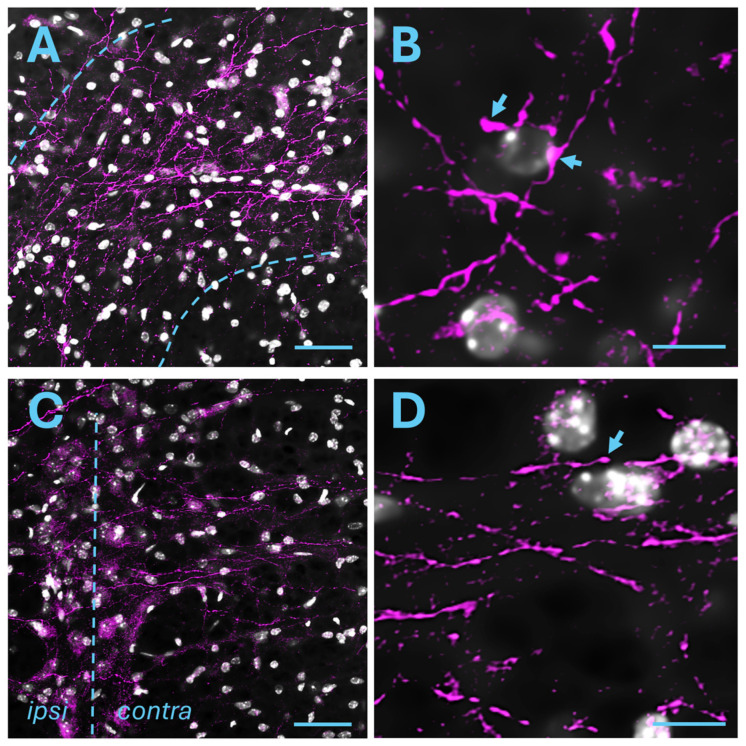
Posterior insular projections to the RMg. (**A**) Axonal projections expressing mCherry (magenta) in the ipsilateral RMg. Blue dashed lines delineate the RMg. (**B**) Higher magnification of the image in (**A**), showing axons and varicosities. (**C**) Axonal projections expressing mCherry (magenta) in the ipsilateral RMg. Blue dashed line denotes the midline, dividing the ipsilateral and the contralateral site. Axons crossing from the ipsilateral to the contralateral site are visible. (**D**) Higher magnification of the image in (**C**). Nuclei are stained with DAPI (grey). Scale bars: 50 μm (**A**,**C**), 10 μm (**B**,**D**). Arrows denote varicosities.

**Figure 5 ijms-25-09185-f005:**
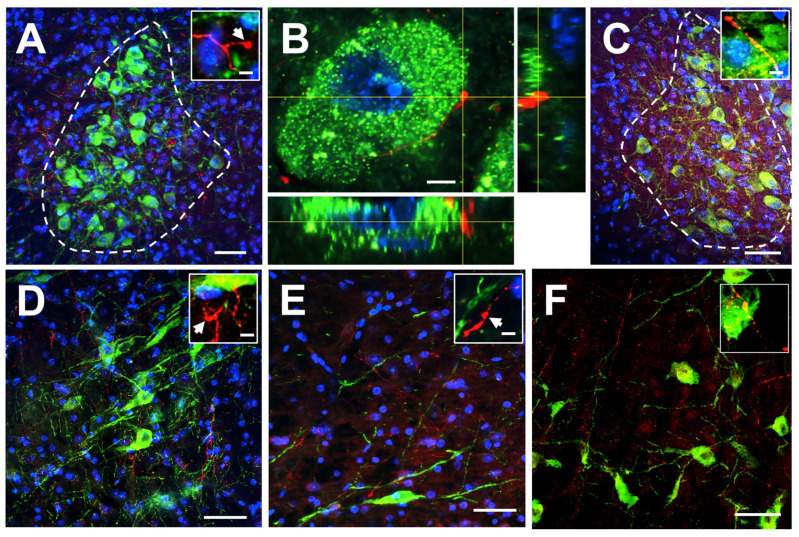
Posterior insular projections to noradrenergic brainstem nuclei. (**A**) Axonal projections from the insula labeled with immunofluorescence for mCherry (red) to the ipsilateral LC. Noradrenergic cells are labelled with TH (green) and nuclei with DAPI (blue; scale bar: 50 μm). The inset shows higher magnification of insula-emanating axons with varicosity-like structures (scale bar: 5 μm). (**B**) Orthogonal view of a high resolution (100×) single imaging plane showing a noradrenergic neuron (green) labelled with anti-TH antibodies and the respective x–z, y–z planes reconstructed from a 3D stack. Varicosities (denoted by the crossing of the yellow lines) from insular projections (red) are shown in close contact around the TH-positive cell (DAPI stain is in blue; scale bar: 5 μm). (**C**) Axonal projections from the insula labeled with immunofluorescence for mCherry (red) to the contralateral LC. Noradrenergic cells are labelled with TH (green) and nuclei with DAPI (blue; scale bar: 50 μm). The inset shows higher magnification of insula-emanating axons with varicosity-like structures (scale bar: 5 μm). (**D**) Axonal projections from the insula labeled with immunofluorescence for mCherry (red) to the ipsilateral SubCD. Noradrenergic cells are labelled with TH (green) and nuclei with DAPI (blue; scale bar: 50 μm). The inset shows higher magnification of insula-emanating axons with varicosity-like structure (scale bar: 5 μm). (**E**) Axonal projections from the insula labeled with immunofluorescence for mCherry (red) to the ipsilateral A5 nucleus. Noradrenergic cells are labelled with TH (green) and nuclei with DAPI (blue; scale bar: 50 μm). The inset shows higher magnification of insula-emanating axons with varicosity-like structures (scale bar: 5 μm). (**F**) Axonal projections from the insula labeled with immunofluorescence for mCherry (red) to the ipsilateral A7 nucleus. Noradrenergic cells are labelled with TH (green) (scale bar: 50 μm). The inset shows higher magnification of insula-emanating axons with varicosity-like structures (scale bar: 5 μm). Arrows denote varicosities.

**Figure 6 ijms-25-09185-f006:**
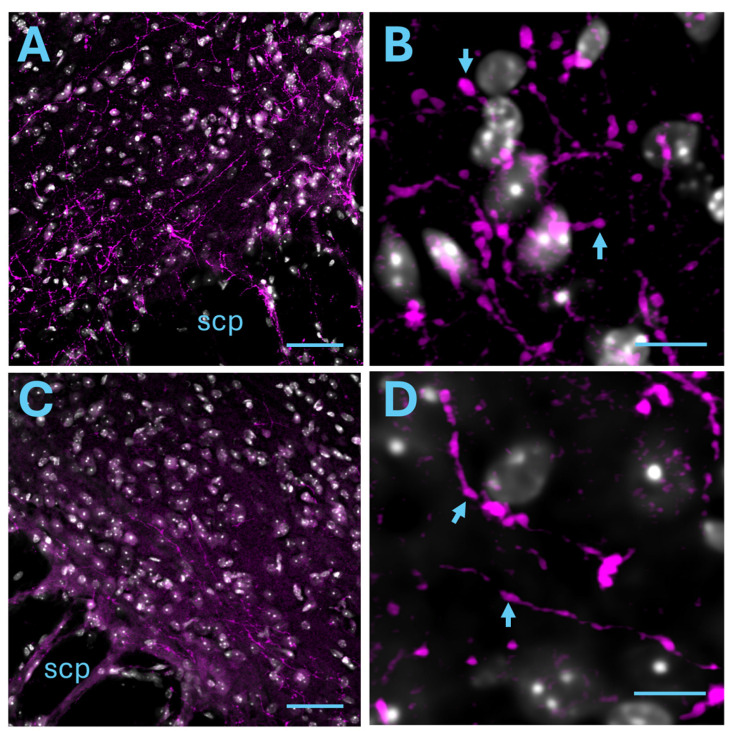
Posterior insular projections to the PBN. (**A**) Insular axonal projections to the ipsilateral lPBN immunolabeled for mCherry (magenta). (**B**) Higher magnification of (**A**). Insula en-passant axonal varicosities are visible. (**C**) Insular axonal projections to the contralateral lPBN immunolabeled for mCherry (magenta). (**D**) Higher magnification of (**A**). Insula en-passant axonal varicosities are visible. scp: Superior Cerebellar Peduncle, grey: DAPI staining, scale bar: 50 μm (**A**,**C**), 10 μm (**B**,**D**). Arrows denote varicosities.

**Figure 7 ijms-25-09185-f007:**
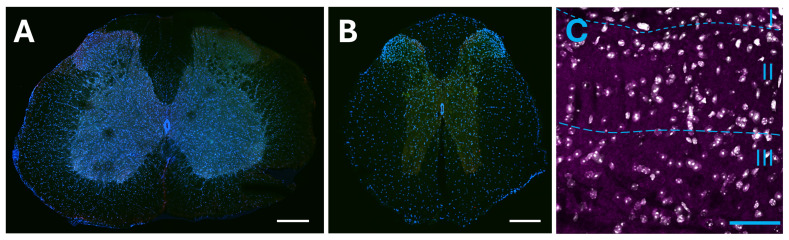
Lack of axonal projection from the insula to the spinal dorsal horn projections. (**A**) Lumbar section of the spinal cord stained with anti-mCherry (red) and with DAPI (blue); green autofluorescence was used for anatomical feature recognition (scale bar: 250 μm). (**B**) Thoracic level of the spinal cord stained with anti-mCherry (red) and with DAPI (blue); green autofluorescence was used for anatomical feature recognition (scale bar: 250 μm). (**C**) Higher resolution of the contralateral superficial lumbar dorsal horn. In grey, the nuclear staining is shown (DAPI), while in magenta, the anti-mCherry staining for insular axons is imaged. No evidence of axonal shafts or varicosities was identified. Dashed lines denote the borders between different dorsal horn laminas (I, II, III; scale bar: 50 μm).

**Figure 8 ijms-25-09185-f008:**
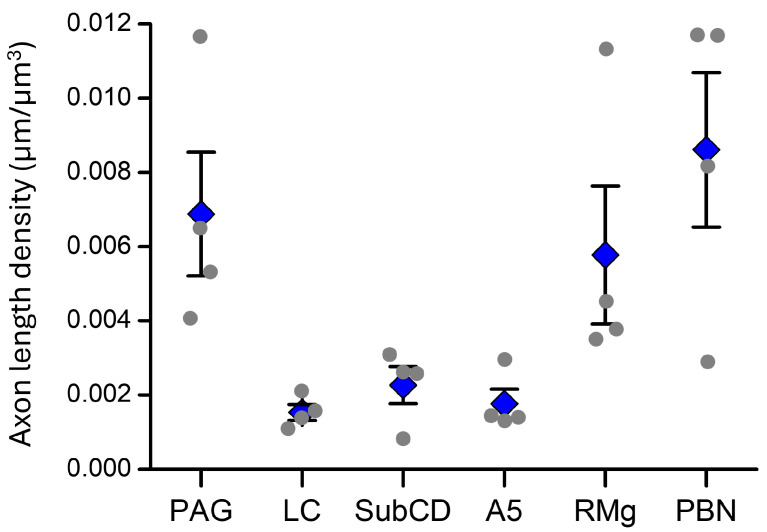
Comparison of axonal length densities for insular projections to different regions of the brainstem. Blue rhombi: means ± SEM. Grey circles: individual values for the different mice. PAG: ventrolateral periaqueductal grey, RMg: nucleus raphe magnus, LC: locus coeruleus, SubCD: dorsal subcoeruleus nucleus, A5: noradrenergic nucleus A5, PBN: lateral parabrachial nucleus. Axonal densities between the brainstem nuclei differed statistically (rmANOVA, F(5, 15) = 7.71, *p* = 0.019).

## Data Availability

Data is contained within the article or Appendix A.

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
