# Peer review of "Multiple Posterior Insula Projections to the Brainstem Descending Pain Modulatory System"

_ijms, 2024, doi:10.3390/ijms25179185_

Round 1

Reviewer 1 Report

Comments and Suggestions for Authors

The article outlines a small study to determine the presence of axonal projections from the posterior insular cortex to regions to the descending pain modulatory system (DPMS). tradditonal tracing between these regions has been done previously, thus the work is largely confirmatory, the novelty is, perhaps, the use of viral vectors that would offer follow up functional assessment of the pathways.

I have some comments and suggestions that are intended to improve the manuscript.

Introduction – the introduction is lacking in specific information about the insular cortex and its known anatomical connectivity to regions of the DPMS that have been found by previous work. The reader should understand what has been found previously, what the rationale for this current study was and why it will address a knowledge gap and/or build capacity for further work.

There should be some introduction of viral vector tracing, how it compares to traditional methods and why the authors chose this approach. It could be made clearer to the reader that using a CaMKII promotor will selectively transfect glutaminergic projections, why the authors chose this promotor as opposed to a non-selective promotor and how this related to what is known about the neurochemistry of projections from the posterior insular cortex.

To many words are given to explain the functional roles of regions of the DPMS that are not directly relevant to research carried out in the manuscript. This information would be better presented in the discussion to provide a broader context to interpret the findings.

Methods –  

Viral vector –  AAV8 vectors can travel both anterograde and retrograde (https://www.ncbi.nlm.nih.gov/pmc/articles/PMC9642939/), albeit not in equal proportion. There is a fair chance that you will find neuronal cell bodies in the regions you are tracing too. If so, this would confound interpretation of the source of these processes. You should make clear the presence or not of neuronal cell bodies in downstream regions of the DPMS. The implications of using AAV8 should be outlined in the discussion.

There is a lack of specific detail on the image analysis process used to quantify projections in the regions of the DPMS. From the information presented, neuronal process were manually traced from two 15um stacked images that were overlayed. The scarcity of criteria for, say, identification of positive labelling, minimum process length that was included in the analysis etc, would make it impossible to reproduce the method. This is the very purpose of the methods section and must be improved.

Its not clear why ImageJ processes for thresholding and background subtraction were not conducted that would have allowed for more sensitive detection of positive staining and provided parameter values that could have been quoted to improve reproducibility. Moreover, manually detecting and tracing positive staining would have likely been majorly impacted by computer monitor resolution and lighting conditions in the room at the time of analysis. I am concerned that details have been missed due to the manual approach used.

There is no detail of how spread of vector expression was detected, quantified and combined to provide composite image in figure 1

Experimental numbers – how many animals were excluded from the study?

Results –

Given the low number of animals used, individualized data should be presented to provide the reader good understanding of the variability.  

1)        Injection site– the tracing of projections will be determined by the delivery of tracer. The authors present a group average spread but this does not describe the variability between animals adequately. Individual animals should be presented. Moreover, spread of the vector does not adequately indicate the location of vector delivery as there will be tracing of neuronal processes that course within the structure. Hamilton syringes should leave a sizable tract lesion that should allow the authors to locate the site of vector delivery. It is this that will critically determine the anatomical connectivity of the region. This should be presented for each animal.

2)        Page 2 line 85. The authors quote “68-90 cell bodies” but this is impossible to interpret without more information. Please quote a group mean value and variability and provide information in the methods of how this number was obtained (e.g. counted in every serial section across the posterior insular? Or from 1 section in 3 - and therefore an underestimation, etc) 

3)        data points in figure 7 should be individualized.

4)        Meso-scale location of axonal processes within DPMS structures. The authors only really confirm that there were some positively labelled processes within their target structures with some qualitative statements of the broader pattern of tracing. These statements could and should be supported with evidence. Using ImageJ to detect positive staining and removing the background will likely show the presence of neuronal processes throughout the structures and these mesoscale images could be presented and quantified. This would provide additional useful information for the reader.

Retrograde tracing from the PAG should be included in the main article. It is highly relevant to the interpretation of findings and there appears no need for it to be supplementary.

The authors state the presence of axonal projections and varicosities throughout the results but do not indicate what they consider these to be. Pointing these out in the images would be helpful. Additionally, stating the relevance of their presence to the aims of the study in the discussion would also be useful for the reader.

Some minor comments on English

Page 1 line 29 Biphasic. This implies a temporal quality that does not make sense to me. Perhaps, bi-directional?

Author Response

We thank the referee for his constructive and extensive review. While, indeed, part of the study confirms in mice previous work done in rat that will aid future functional studies, another part of this work supplements the previous part with novel findings on the insula connectivity to other noradrenergic nuclei, besides the LC, and which has not been studied before. In addition, we provide experiments testing the direct connectivity of the insula to the spinal cord. Albeit, a negative result, we believe it is an important piece information for the readers, as definitive knowledge for such a connection is not available yet. Below our responses to the referee’s comments

Comments 1: Introduction – the introduction is lacking in specific information about the insular cortex and its known anatomical connectivity to regions of the DPMS that have been found by previous work. The reader should understand what has been found previously, what the rationale for this current study was and why it will address a knowledge gap and/or build capacity for further work.

Response 1: Indeed, the introduction lacked information on previously published insula to DPMS connectivity. We added information to the previous work and linked it to the rational of this study (lines 68-76, page 2).  

Comments 2: There should be some introduction of viral vector tracing, how it compares to traditional methods and why the authors chose this approach. It could be made clearer to the reader that using a CaMKII promotor will selectively transfect glutaminergic projections, why the authors chose this promotor as opposed to a non-selective promotor and how this related to what is known about the neurochemistry of projections from the posterior insular cortex.

Response 2: We agree. We added information both in the introduction (lines 76-77, page 2) and the discussion to make the reader aware of the glutamatergic nature of the axonal projections we trace.  We discuss now the comparison between viral and traditional tracing methods (lines 357-370, page 10).   

Comments 3: Viral vector –  AAV8 vectors can travel both anterograde and retrograde (https://www.ncbi.nlm.nih.gov/pmc/articles/PMC9642939/), albeit not in equal proportion. There is a fair chance that you will find neuronal cell bodies in the regions you are tracing too. If so, this would confound interpretation of the source of these processes. You should make clear the presence or not of neuronal cell bodies in downstream regions of the DPMS. The implications of using AAV8 should be outlined in the discussion.

Response 3: We routinely check for retrograde infection in the contralateral insula, the primary somatosensory cortex, the mid- and anterior- cingulate cortex and the amygdala and never found cells expressing mCherry in the concentrations used in this study. Only exception were two single cells (one in the contralateral insula and one in the ipsilateral primary somatosensory cortex) each in two different animals (out of 8), in which the virus was used undiluted. We also did not detect any retrograde infection in the brainstem target regions (Some, faintly visible cells in different DPMS nuclei, are just background/autofluorescence, that are also visible in immunostainings from control animals that have not been injected with viruses). AAV8 is being used and published widely and retrograde infections are not reported often. We now discuss the possibility of retrograde transport in the manuscript and reference the study by Masamizu et al. 2011, that shows retrograde AAV8 transport in primates (lines 371-377, page 11).   In addition, we would like to point out that in two of the studies (we did not check all) referenced by Surdyka and Figiel as showing AAV8 retrograde transport, do not actually show retrograde transport of AAV8 itself. Both studies (Pina and Cunningham, 2017 – doi: 10.1016/j.nlm.2016.11.013; Parfitt et all., 2017 – doi: 10.1038/npp.2017.56) use retrograde capable helper viruses (HSV-cre & CAV-cre) to express the AAV8 cargo in the target regions (with the AAV8 injected in the target region).

Comments 4: There is a lack of specific detail on the image analysis process used to quantify projections in the regions of the DPMS. From the information presented, neuronal process were manually traced from two 15um stacked images that were overlayed. The scarcity of criteria for, say, identification of positive labelling, minimum process length that was included in the analysis etc, would make it impossible to reproduce the method. This is the very purpose of the methods section and must be improved.

Response 4: The image analysis process was indeed not well presented. The z-stacks that were summed, contained 25 images and not two. These 25 images transversed/covered a tissue thickness of 15μm deep. We improved this part in the methods section (lines 443-447, page 12).

Comments 5: Its not clear why ImageJ processes for thresholding and background subtraction were not conducted that would have allowed for more sensitive detection of positive staining and provided parameter values that could have been quoted to improve reproducibility. Moreover, manually detecting and tracing positive staining would have likely been majorly impacted by computer monitor resolution and lighting conditions in the room at the time of analysis. I am concerned that details have been missed due to the manual approach used.

Response 5: Indeed, we initially intended to automate the detection and measuring of axonal processes in ImageJ, instead of the time-consuming manual approach we later chose. However, thresholding and background subtraction also influence the fluorescent staining, with the thin axonal processes being more susceptible. Because we observed that the different areas studied displayed different backgrounds and autofluorescence (even in the same slice) we were concerned that thresholding and background subtraction would differentially affect axonal length measurements and distort the comparison between regions (this comparison was one of the motivations for the study, and one to influence our future functional experiments). Although, not without its drawbacks, we believe that with the manual approach we avoided biasing the results for the different DPMS nuclei.      

Comments 6: There is no detail of how spread of vector expression was detected, quantified and combined to provide composite image in figure 1

Response 6: We agree. We have accordingly added more details on our assessment of  mCherry expression spread in the methods section (lines 433-442, page 12).

Comments 7: Experimental numbers – how many animals were excluded from the study?

Response 7: Thanks for pointing that out. We added information on the excluded animals in the results section (lines 87-90, page 2).

 Comments 8: Injection site– the tracing of projections will be determined by the delivery of tracer. The authors present a group average spread but this does not describe the variability between animals adequately. Individual animals should be presented. Moreover, spread of the vector does not adequately indicate the location of vector delivery as there will be tracing of neuronal processes that course within the structure. Hamilton syringes should leave a sizable tract lesion that should allow the authors to locate the site of vector delivery. It is this that will critically determine the anatomical connectivity of the region. This should be presented for each animal.

Response 8: Indeed, the spread of mCherry expression does not adequately indicate the spread of transfection, as mCherry is expressed in the dendritic tree and in distal axons. Therefore, we included a more quantitative approach for the rostro-caudal extent of transfection by counting the number of cells expressing mCherry at different rostro-caudal levels (see below, Fig. S5). Accordingly, we added figures with mCherry expression patterns for individual animals (Supplementary figures S1-S4). We also show images of mCherry expression at the site of injection for each animal (Fig. S1-4).     

Comments 9: Page 2 line 85. The authors quote “68-90 cell bodies” but this is impossible to interpret without more information. Please quote a group mean value and variability and provide information in the methods of how this number was obtained (e.g. counted in every serial section across the posterior insular? Or from 1 section in 3 - and therefore an underestimation, etc) 

Response 9: We agree. The expression of mCherry diffuses from the site of vector uptake to dendrites and to extensively branched axons. Added to that, our arbitrary estimation of spread does not add quantitative value. We therefore include now a quantitative measure of viral transfection, i.e. the number of mCherry positive somata at different levels of the rostro-caudal axis. We plot the mean values from sections at specific locations spanning the rostro-caudal axis for all animals (Supplementary figure S5).   

Comments 10: data points in figure 7 should be individualized.

Response 10: Individual data points were added in Figure 7 (Figure 8 now, line 325, page 10)

Comments 11: Meso-scale location of axonal processes within DPMS structures. The authors only really confirm that there were some positively labelled processes within their target structures with some qualitative statements of the broader pattern of tracing. These statements could and should be supported with evidence. Using ImageJ to detect positive staining and removing the background will likely show the presence of neuronal processes throughout the structures and these mesoscale images could be presented and quantified. This would provide additional useful information for the reader

Response 11: We added mesoscale images of different DPMS areas (PAG, Rostral Ventomedial Medulla, LC, SubCD, PBN). Because of the lower magnification used for collecting the images, as well as the thinness of the processes and the effects of background subtracting/thresholding, the qualities of the images are not adequate to our opinion. Therefore, we chose to present these in the supplementary section, allowing for the reader to inspect larger images (Supplementary figures S6-8).    

Comments 12: Retrograde tracing from the PAG should be included in the main article. It is highly relevant to the interpretation of findings and there appears no need for it to be supplementary

Response 12: We included the retrograde tracing figure in the main article as suggested (figure 3, lines 120-122, 164-173). We also added images from another retrograde tracing experiment.

Comments 13: The authors state the presence of axonal projections and varicosities throughout the results but do not indicate what they consider these to be. Pointing these out in the images would be helpful. Additionally, stating the relevance of their presence to the aims of the study in the discussion would also be useful for the reader.

Response 13: We added arrows pointing to the varicosities in several images (Figures 2B/D/F, 4B/D, 5A/B/D/E, 6B/D). The relevance of varicosities is discussed in lines 338-341 (page 10).

Comments 14: Page 1 line 29 Biphasic. This implies a temporal quality that does not make sense to me. Perhaps, bi-directional?

Response 14: Indeed, bidirectional is the correct term. Changed it in line 29 (page1).

Reviewer 2 Report

Comments and Suggestions for Authors

The manuscript provides an anatomical analysis of neuronal circuits that are relevant for pain control. By using viral vectors the authors show multiple projections from the posterior insular cortex to several brainstem nuclei, that are relevant in descending pain modulation. The manuscript is interesting.

Please explain better the circuit described in lines 47-48, namely in what concerns “might directly gate the descending control system”.

In line 101, section Results, the sentence “The PAG displays a functional organization into longitudinal columns [41]” seems out of place. Please consider deletion

Please consider an explanation about the reasons to use CTB, lines 110-112, as IJMS is not a neuroscience journal.

As in the comment above, the sentence in lines 115-116, is not adequate for the results The RMg is part of the rostral ventromedial medulla, an area implicated in de- 115 scending pain modulation.

Please explain better the methods to clearly separate the LC from the subcoeruleus, as this is frequently difficult.

In lines 228-229, at the beginning of the Discussion, the authors state that they “used low-titre viral injections to contain and restrict spread within the posterior insula cortex”. This is better at the end of the discussion. Nevertheless, the authors should discuss if there are data that demonstrate that the injections were really restricted, namely in what concerns the patter of projections of surrounding regions.

Besides the anatomical questions of the possible redundance of multiple projections, that the authors use to finish the discussion, do the authors consider that functional evaluation of the circuits would be important

Immunofluorescence: Can the authors cite studies that support the specificity of the primary antibodies used?

Author Response

We thank the referee for his constructive review. Please find bellow our responses

Comments 1. Please explain better the circuit described in lines 47-48, namely in what concerns “might directly gate the descending control system”.

Response 1: A better explanation has been added for the circuit in question (48-50, page 2).

Comments 2. In line 101, section Results, the sentence “The PAG displays a functional organization into longitudinal columns [41]” seems out of place. Please consider deletion

Response 2: We changed the text accordingly (deleted, line 112).

Comments 3. Please consider an explanation about the reasons to use CTB, lines 110-112, as IJMS is not a neuroscience journal.

Response 3: Thanks for point this out. We added the reason for implementing the CTB experiments in the results section (line 120-122, page 3).

Comments 4. As in the comment above, the sentence in lines 115-116, is not adequate for the results The RMg is part of the rostral ventromedial medulla, an area implicated in de- 115 scending pain modulation.

Response 4: We changed the text accordingly (line 127, page 3).

Comments 5. Please explain better the methods to clearly separate the LC from the subcoeruleus, as this is frequently difficult.

Response 5: We updated the methods relating to LC and SubCD delineation line 425-430, page 12). We also provide now an image (Fig S8) as a visual guide of the location of SubCD we used for our analysis in relation to other brainstem areas.

Comments 6. In lines 228-229, at the beginning of the Discussion, the authors state that they “used low-titre viral injections to contain and restrict spread within the posterior insula cortex”. This is better at the end of the discussion. Nevertheless, the authors should discuss if there are data that demonstrate that the injections were really restricted, namely in what concerns the patter of projections of surrounding regions.

Response 6: We discuss the specificity of the insula DPMS connection in lines 337-341 (page 1)  and added mesoscale images (Supplementary figures S6-8) showing expression patterns to the surrounding regions. We also moved the discussion of low titre injections to the end of the discussion, together with other methodological considerations. 

Comments 7. Besides the anatomical questions of the possible redundance of multiple projections, that the authors use to finish the discussion, do the authors consider that functional evaluation of the circuits would be important

Response 7: Indeed, we do consider the functional evaluation of importance. The rational for this work is our future functional experiments. We added a relevant statement at the end of the discussion (lines 381-383 page 11)

Comments 8. Immunofluorescence: Can the authors cite studies that support the specificity of the primary antibodies used?

Response 8: The anti-TH antibodies are well known and used in many studies and their specificity can be deduced from studies involving degeneration of dopaminergic and noradrenergic neurons (10.1038/s41598-024-65735-5 & 10.1016/j.ejphar.2024.176573). On the other hand, we are not aware on any published work that supports the specificity of the anti-mCherry antibody. However, this work offers plenty of evidence on its specificity: 1) Expression of mCherry is restricted only around the injection sites (Fig. 1, S1-4). 2) Axonal expression of mCherry is found only in certain areas (for example not found with the spinal cord at all) and at different densities (Fig 7, 8, S6-8). 3) No axonal immunofluorescence is seen in brains from animals that have not been injected with mCherry viral vectors (not shown).  

Round 2

Reviewer 1 Report

Comments and Suggestions for Authors

the additions to the manuscript have improved the manuscript to a satisfactory standard. I congratulate the authors on their hard work.